# Adenosinergic System and Neuroendocrine Syncope: What Is the Link?

**DOI:** 10.3390/cells12162027

**Published:** 2023-08-08

**Authors:** Régis Guieu, Julien Fromonot, Giovanna Mottola, Baptiste Maille, Marion Marlinge, Antonella Groppelli, Samantha Conte, Yassina Bechah, Nathalie Lalevee, Pierre Michelet, Mohamed Hamdan, Michele Brignole, Jean Claude Deharo

**Affiliations:** 1Centre for Cardiovascular Research and Nutrition (C2VN), INSERM, INRAE, AMU, 13005 Marseille, France; julien.fromonot@univ-amu.fr (J.F.); bpatiste.maille@ap-hm.fr (B.M.); marion.marlinge@ap-hm.fr (M.M.); samantha.conte@univ-amu.fr (S.C.); natahlie.lalevee@univ-amu.fr (N.L.); pierre.michelet@ap-hm.fr (P.M.); jean-claude.deharo@ap-hm.fr (J.C.D.); 2Laboratory of Biochemistry, AP-HM, 13005 Marseille, France; yassina.bechah@ap-hm.fr; 3Department of Cardiology, Syncope Unit, Timone Hospital, 13005 Marseille, France; 4IRCCS Istituto Auxologico Italiano, Department of Cardiology, San Luca Hospital, Piazzale Brescia 20, 20149 Milan, Italy; antonella.groppelli@unimib.it (A.G.); mbrignole@outlook.it (M.B.); 5Department of Anesthesia and Reanimation, Hopital Conception, 13005 Marseille, France; 6Department of Medicine, Division of Cardiovascular Medicine, University of Wisconsin School of Medicine and Public Health, Madison, WI 53705, USA; mhamdanhrs@gmail.com

**Keywords:** neuroendocrine syncope, adenosine, ionic channels, receptor reserve

## Abstract

Although very common, the precise mechanisms that explain the symptomatology of neuroendocrine syncope (NES) remain poorly understood. This disease, which can be very incapacitating, manifests itself as a drop in blood pressure secondary to vasodilation and/or extreme slowing of heart rate. As studies continue, the involvement of the adenosinergic system is becoming increasingly evident. Adenosine, which is an ATP derivative, may be involved in a large number of cases. Adenosine acts on G protein-coupled receptors with seven transmembrane domains. A_1_ and A_2A_ adenosine receptor dysfunction seem to be particularly implicated since the activation leads to severe bradycardia or vasodilation, respectively, two cardinal symptoms of NES. This mini-review aims to shed light on the links between dysfunction of the adenosinergic system and NHS. In particular, signal transduction pathways through the modulation of cAMP production and ion channels in relation to effects on the cardiovascular system are addressed. A better understanding of these mechanisms could guide the pharmacological development of new therapeutic approaches.

## 1. Introduction

Neuroendocrine Syncopes (NES) are frequent in the general population. NES is characterized by a partial or total loss of consciousness due to a drop in systolic blood pressure [1]. The fall in blood pressure is secondary to vasoplegia and/or severe bradycardia while sometimes an atrioventricular block (AVB) may occur [2]. In a large number of cases, loss of consciousness is preceded by prodromes like dizziness, nausea, abdominal pain or cephalalgia. Less frequently, the loss of consciousness occurs without prodromes (sudden syncope) [3,4]. NHS account for 3 to 5% of emergency entrance and 1% of hospitalization with a mortality of 0.28% [5,6,7].

The recurrence occurs in 35% of cases while discomfort is followed by injuries in 29% of cases [8]. Two tests are commonly used for the exploration of these syncopes: the head-up tilt test (HUT) and the intravenous bolus injection of ATP or adenosine [1]. Both tests reproduce the symptomatology that occurs in the daily life of patients. Although it is a very common condition, physiological and molecular mechanisms of NES remained unclear, which makes the delivery of appropriate treatment a challenge. However, it is important to know the cause(s) of clinical manifestations at both physiological and molecular levels, in order to be able to deliver an appropriate treatment. In recent years, it has become clear that the cause of symptoms may be related to a dysfunction of the adenosinergic system. This is supported by the fact that (1) administration of adenosine reproduces the symptomatology of NES [9,10]; (2) during the head up-tilt test, a maneuver that reproduces the symptoms, an increase in adenosine plasma level (APL) was observed and was associated with the drop blood pressure [11] and (3) antagonists of adenosine receptors are efficient in the treatment of some kind of NES [12,13]. In fact, adenosine, an ATP derivative, exerts a powerful cardio-inhibition and vasodilation via activation of its receptors, two phenomena which are found during the clinical manifestations of NES. The aim of this review is to highlight the latest data on the possible role of the adenosinergic system in NES.

## 2. Source and Metabolism of Adenosine

Adenosine is a ubiquitous nucleoside, present in most cells but particularly in endothelial and muscle cells, which mainly comes from the dephosphorylation of ATP via intra-cellular nucleotidases or via ecto-nucleotidase CD39 and CD73 (Figure 1). The methionine cycle represents only a small part of the adenosine source and does not allow rapid adenosine release, unlike ATP dephosphorylation. The ATP pathway is particularly active when there is cellular or tissue stress, such as during inflammation, hypoxia or ischemia. At the intra-cellular level, adenosine can be rephosphorylated in adenine nucleotides by kinases, but this rephosphorylation process is inhibited by the hypoxia-inducible factor (HIFa) in conditions of low oxygen tension, leading to a strong release of adenosine in the extra-cellular spaces via the equilibrative nucleoside transporter (ENT1).

Adenosine is also deaminated into inosine by intra-cellular or plasmatic ADA or by ADA linked to mononuclear cells (MCADA (Figure 1), the final product being uric acid. 

Adenosine mainly comes from the dephosphorylation of adenyl nucleotides both at the intra and extracellular levels. The dephosphorylation occurs via nucleotidases in intracellular spaces and via CD39 and CD73 at the extracellular level. Part of intra-cellular adenosine is rephosphorylate via adenosine kinases. A short part of adenosine comes from the methionine cycle.

The behavior of adenosine is trice. (1) adenosine is quickly deaminated into inosine by an intra and extracellular adenosine deaminase (ADA). Adenosine is also deaminated via MCADA (mononuclear cell adenosine deaminase) which is a plasmatic ADA linked to the CD26 transmembrane protein of the mononuclear cells via non-covalent binding. The final product is uric acid after the action of xanthine oxydase (XO). (2) Extracellular adenosine is quickly uptaken by red blood cells via the equilibrative nucleoside transporter (mainly ENT1). (3) Part of adenosine that has not been deaminated or uptaken by red blood cells acts on three subtypes of receptors named A_1_ R, A_2_ AR, A_2B_ R and A_3_ R with a strong impact on the cardiovascular system.

ATP: adenosine triphosphate; ADP: adenosine diphosphate; AMP: adenosine monophosphate. SAHH: S-adenosyl-homocysteine hydrolase.

Despite its short half-life of adenosine in the extracellular space (mainly due to uptake by red blood cells and deamination by ADA), adenosine strongly impacts the cardiovascular system via its receptors.

## 3. Adenosine Receptors Implicated in Blood Pressure and Heart Rate

Adenosine acts on the cardiovascular system via its membrane receptors namely A_1_ R, A_2A_ R, A_2B_ R and A_3_ R, pending on their primary sequence and their action on adenylcyclase pathway [14,15]. The activation of A_1_ R or A_3_ R leads to the inhibition of adenylcyclase and thus to the decrease in cAMP production, while conversely, the activation of A_2_ R subgroups leads to cAMP production.

A_1_ R is expressed in the sinus node and atrio-ventricular node where they control heart rate and AV conduction. The main effects of A_1_ R activation are the “direct” (cAMP-independent) activation of the inwardly rectifying K+ current IKAdo, Ach [16]. These channels are also activated by Ach. The direct effect occurs via the protein–protein interaction of the G-protein (probably the beta/gamma dimere complex [17]). The activation of these inward rectifying potassium channels leads to hyperpolarization of the synaptic membrane resulting in a decreased excitability and slowing of the heart rate with sinus arrest or atrio-ventricular block. The activation of A_1_ R also leads to phospholipase C stimulation (PLC) and thus activates the IP3 cascade, modulating the calcium release from its reticulum storage via the IP3 receptor (Figure 2). Finally, A_1_ R activation leads to the inhibition of adenylcylase activity and thus inhibiting the cAMP production and beyond the cAMP-dependent neurotransmission [18].

On target cells, activation of A_1_ R leads to the activation of the alpha i subunit of the G protein, leading to the inhibition of the production of cAMP (indirect effects). Activation of A_1_ R also leads to the activation of potassium channels, mainly the inward rectifier (IK ado/Ach) and the inhibition of L-type voltage gate calcium channels (Belardinelli 1995).

A_2A_ R is strongly expressed in vessels where they are implicated in the blood flow via smooth cell relaxation. The activation of A_2A_ R leads to the stimulation of adenylcyclase activity and thus the cAMP production and the activation of a protein kinase cAMP-dependent (PKA) (Figure 3). The three consequences of the PKA activation are: (i) activation of K_ATP_ and K_V_ channels; (ii) the inhibition of L-type voltage-gated calcium channels; and (iii) the activation of NO synthase leading to NO release. These effects are all consistent with a vasodilatory effect. Note that the activation of A_2B_ R leads to the same vasodilatory effects but the implication of these receptors in NHS while likely probable is not yet demonstrated especially since their activation needs a high concentration of adenosine [19].

On target cells, activation of A_2A_ R leads to the activation of the alpha S subunit of the G protein, leading to the production of cAMP through the activation of adenylcyclase (AC).

cAMP increase leads to the activation of a protein kinase cAMP-dependent (PKA) with phosphorylates potassium channels (mainly K_ATP_ and K_V_) and also to the inhibition of voltage-sensitive calcium channels (mainly L Type channels). Finally, the activation of A_2A_ R leads to the activation of the enzyme NO synthase (eNOS) via the PKA pathway and thus to NO release. The main effects due to A_2A_ R activation through these signal transduction pathways is vasodilation.

Figure 4 summarizes the main consequences of activation of adenosine receptors on the cardiovascular system in an integrative point of view. 

## 4. Role of the Adenosinergic System in NES

### 4.1. Adenosinergic Profile in NHS Patients

The role of adenosine in NES has long been discussed [4,20]. Thus, there is evidence that an adenosinergic disturbance is implicated in NES. In vasovagal syncope: high basal APL was reported with a significant increase during the HUT associated with fainting [11], while an increase in APL was also reported during HUT [11,21,22]. This increase may explain the lack of prodromes in this patient population. The APL increase was much higher in patients with predominant vasodilation [21]. This is in agreement with the fact that the K_D_ value for A_2A_ R is much higher (1.8 µM, [23]) than for A_1_ R (0.8 µM, [24]). Indeed, in the case of high basal APL, A_1_ R is mostly desensitized and only A_2A_ R could be activated [25]. The increase in basal APL may be explained by the decrease in ADA activity [21]. Furthermore, high A_2A_ R expression was reported in this patient population [26,27] while a specific SNP (c 1083 C > T) in the second exon of the gene encoding A_2A_ R was found [26] but not confirmed by others [27]. In some kinds of syncope, low APL has been described. These forms are most often associated with the absence of prodromes. In these forms, the drop in blood pressure is often associated with severe bradycardia or AVB rather than vasodilation, suggesting that activation of A_1_ R is predominantly at play [3,28,29]. 

The cause of high APL in VVS patients could be explained by a vicious circle (see Figure 5). At the beginning, a possible genetic predisposition with the presence of the specific CC (SNP) although a silent polymorphism, is associated with overproduction of A_2A_ R [26]. The activation of A_2A_ R, via adenosine leads to the overexpression of ENT at the membrane of the PBMCs, triggering the release of adenosine in blood, and thus creating a vicious circle.

CC variant from the second exon of the gene encoding A_2A_ R is associated with an increase in mRNA production followed by an increase in A_2A_ R production. Adenosine acts on A_2A_ R leading to an increase in the number of equilibrative nucleoside transporters (ENT) that control the outflow of adenosine in the extracellular spaces. PKA: protein kinas A; AC: adenylcyclase; GTP: guanosine triphosphates; ATP: adenosine triphosphates; cAMP: cyclic adenosine monophosphate; ENT1: equilibrative nucleoside transporter 1.

Furthermore, a decrease in ADA activity was found in VVS patients with positive HUT [30] which could also contribute to the high APL. The cause of low adenosine in some kinds of NHS remains unknown. Possible ENT1 abnormalities cannot be ruled out as well as abnormal plasma or MCADA activity. However, these hypotheses need further investigation.

### 4.2. Effects of Exogenous Adenosine

ATP injection was used to explore unexplained syncope for a quarter of a century [9,10,31]. Exogenous ATP or adenosine reproduces symptomatology in the subgroup of patients suffering from NES. Positive HUT is associated with an increase in endogenous APL while ATP/adenosine injection reproduces the symptoms in a subgroup of NES patients [1,28]. However, while a small portion of patients exhibit both positive tests, it seems that HUT and ATP tests identify two distinct populations [28,32]. In positive HUT the basal APL is high which increases further during HUT, this increase activates mainly A_2A_ R. In patients with very low basal APL, the adenosine release that occurs during HUT is not sufficient to recruit even high-affinity adenosine receptors, whereas a rapid bolus injection is likely to produce sufficient adenosine concentration to activate at least the high-affinity receptors (mainly A_1_ R) inducing sinus bradycardia and AVB.

### 4.3. The Question of Adenosine Measurement

Numerous methods have been tried to measure adenosine in blood with varying degrees of success, including high-performance liquid chromatography [33], amperometry [34], and mass spectrometry LC-MS/MS [34,35,36]. The issue is not so much the assay method as the sampling conditions. The half-life of adenosine is relatively short in blood but longer in a sample tube. Amperometry, which measures adenosine in real time and thus permits kinetic studies, has shown that the half-life of adenosine at room temperature is of the order of 45 s, longer than in circulating blood (unpublished data). A comparative study a showed good correlation between high-performance liquid chromatography, amperometry and mass spectrometry (LC-MS/MS), the latter being the gold standard [34]. Adenosine determination after blood collection on blotting paper is convenient because it does not require the use of a stop solution, unlike tube sampling. It also allows easy fingertip capillary sampling and samples can be mailed. Finally, the use of blot paper followed by LC-MS/MS appears to be reliable and useful to the dosage of adenosine in blood.

### 4.4. Effects of Dioxygene on the Adenosinergic System

Hypoxia is associated with an increase in intracellular adenosine, (via the HIF pathway see 2.1) with subsequent release in the extra-cellular spaces. Hypoxia also induces an up-regulation of A_2A_ R [36]. Conversely hyperoxia, in animal studies, leads to a drop in APL and a down-regulation of A_2A_ R [37]. This drop is associated with vasoconstriction leading to an increase in blood pressure.

Interestingly in healthy subjects, hyperoxia was shown to increase blood pressure during tilt testing, rendering tilt test negative in subjects with previously positive HUT. These hemodynamic changes were associated with a decrease in APL [38]. This highlights another impact of adenosine in the regulation of blood pressure during HUT via the dioxygene pathway.

## 5. Peripheral Blood Mononuclear Cells (PBMCs) Are a Useful Tool for the Exploration of the Adenosinergic System

PBMCs express on their surface in particular the A_1_ R and A_2A_ R. It has been shown that their level of expression, as well as their function (i.e., cAMP production), mirrored the expression and the function of these same receptors present in the tissues and organs of the cardiovascular system, in particular the myocardium, coronary arteries, aorta and iliac vessels [39,40,41]. PBMCs are permanently in contact with the circulating blood and are therefore subject, like cardiovascular tissues, to variations in the concentration of extracellular adenosine and thus their expression regulation is the same for cardiovascular tissues.

A monoclonal antibody, against the second extra-cellular loop of the human A_2A_ R, which exhibits agonist properties was developed [42]. This unique tool permits us to evaluate A_2A_ R expression on a lot of cardio-vascular tissues, but also PBMCs. This antibody permits also to evaluate in the same time affinity (K_D_) and cAMP production (EC_50_). Using this tool, it was shown that a small fraction of NHS patients exhibits a specific profile of A_2A_ R named receptor reserve or spare receptors [43,44]. 

## 6. The Role of Receptor Reserve (Spare Receptors)

The biological response to the activation of adenosine receptors depends on tissue nature, pathophysiological situations and the nature of the ligand, agonist or antagonist. 

In a general way, the intensity of the biological effects (i.e., cAMP production) correlates with the number of receptors which are activated by the ligand (here adenosine). Furthermore, the cAMP production in target cells correlates with the intensity of physiological effects (heart frequency and/or vasodilation). The concept developed by Clark in the 1930s then Stephenson [45,46] is that in some cases a small fraction of activated receptors is sufficient to obtain maximal biological effects (Figure 6). From a pharmacological point of view, this concept means that the presence of spare receptors can be advocated when the half-maximal biological effects evaluated as an example by the half-maximal cAMP production (EC_50_) is lower than the K_D_ value [23,43,44,47]. Both A_1_ R and A_2A_ R spare receptors have been described in the heart [23,48,49]. While the presence of spare adenosine receptors is usual in healthy small mammalians [23], in humans, their presence seems to be an adaptive mechanism to compensate for low adenosine levels, low adenosine receptor expression or both [47]. Thus, the presence of spare could provide the cell with a survival mechanism in order to be able to maintain a signal transduction process in the event of a shortage of the ligand, shortage of the receptor or both. 

The presence of spare receptors is also a way for tissues to favor one biological effect over another. For example, the presence of A_2A_ R spare in the heart, can promote coronary vasodilation rather than heart rate, the EC_50_ value for vasodilation being lower than the EC_50_ value for the modulation of heart rhythm [23]. Thus, in the event of a shortage of the ligand or receptors, the presence of spare makes it possible to favor a “protective” reaction rather than another to slow down the heart rate to save the myocardium or to dilate vessels to lower blood pressure.

The presence of spare also explains that the response to agonist or antagonist vary depending on pathophysiological situations and tissue nature and finally may differ from one patient to another. The presence of spare A_2A_ R in some kinds of NES has been reported [43,44]. Interestingly, a spare A_2A_ R was found in a group of patients with very low APL [44], and sometimes in the case of low A_2A_ R expression or both (unpublished data). 

In some patients with NES, a very low APL was measured which is 5 to 10 folds under the EC_50_ value, and thus in these conditions, no adenosine receptors could be activated.

For unknown reasons (infra-clinical hypoxia or inflammation), the APL increase over the EC_50_ value leads to sudden maximum biological effects, objectified by a maximum production of cAMP in the target cells (Figure 6 and Figure 7). 

The rapidity of the biological response does not leave room for prodromes. If the presence of spare receptors concerns A_1_ R, the activation of the receptors during APL increase results in severe bradycardia, sinus arrest or AVB. If it concerns A_2A_ R, the clinical manifestations will be sudden vasoplegia (Figure 8). 

## 7. Effects of Adenosine Receptor Antagonists

Xanthine derivatives like caffeine or theophylline are nonspecific adenosine receptor antagonists. Caffeine binds to adenosine receptors with an affinity close to a 1 to 10 micromolar range [50]. Furthermore, caffeine allows the inflow of calcium from the external environment via the opening of TRP channels [51]. In addition, caffeine activates Ryanodine receptors and promotes the outflow of calcium from the storage site in the reticulum. Thus, caffeine or theophylline favors the calcium induce calcium release phenomenon leading to positive inotropy in the myocardium. In vessels, caffeine stimulates sympathetic activity leading to norepinephrine release and vasoconstriction. The administration of caffeine stimulates the juxtaglomerular apparatus and causes the secretion of renin and therefore the release of aldosterone, the reabsorption of sodium by the kidney leading to an increase in robbery and beyond the pressure arterial [52,53]. Finally, caffeine at high concentrations also inhibits phosphodiesterase.

Theophylline’s efficacy in preventing syncope was first demonstrated in a very small number of patients, with an efficacy that seemed superior to cardiac pacing [54]. More recently, due to its effects, theophylline has been successfully used in the prevention of syncope recurrence [12,13]. While theophylline’s initial indication seemed to be reserved for syncope with low adenosine, it now appears to be effective in syncope with high or normal adenosine concentration [12]. Thus, theophylline administration decreases the number of syncope episodes [12], more particularly in patients affected by low adenosine syncope [13]. Apart from the fact that they are quite often poorly supported, adenosine receptor antagonists can be ineffective in the presence of spare receptors. Indeed, in this case, the concentration of the antagonist must be very high in order to displace the adenosine from all the receptors that it occupies (see Figure 9).

## 8. Effects of Pacing

In a few observational case studies performed in patients with “Low adenosine syncope”, cardiac pacing showed to be successful in preventing syncopal recurrences in patients in whom who had ECG documentation of a systolic pause due to sinus arrest or AV block [3]. In a small multicenter trial [55], performed in 80 highly selected elderly patients with unexplained unpredictable syncope who had a positive response to an intravenous injection of a bolus of 20 mg of ATP dual-chamber cardiac pacing, the 2-year syncope recurrence rate was significantly reduced from 69% in the control group to 23% in the active group. For the above reasons, cardiac pacing may be considered to reduce syncope recurrences in patients with the clinical features of adenosine-sensitive syncope according to the most recent ESC guidelines on syncope [56] and on Cardiac pacing [57] (class IIb recommendation).

## 9. Conclusions

While the adenosinergic system appears to be an indispensable control system for the proper functioning of the cardiovascular system, its dysfunction can be the cause of pathologies that can be disabling and spectacular. A better understanding of the molecular mechanisms that lead to clinical manifestations should enable better management of NES.

## Figures and Tables

**Figure 1 cells-12-02027-f001:**
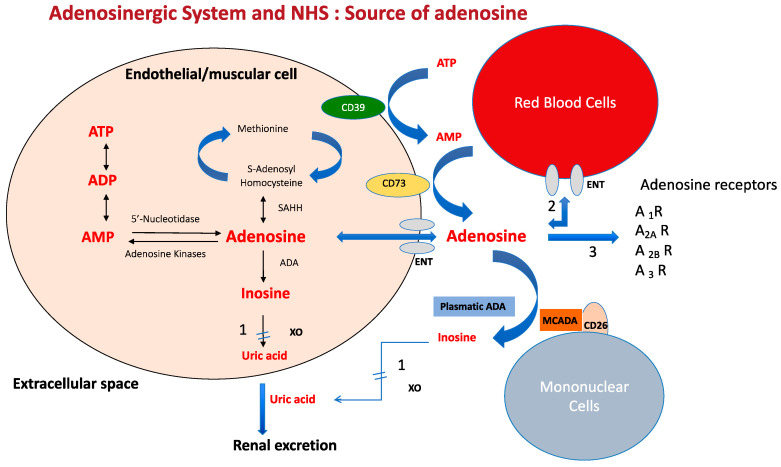
Main source of adenosine.

**Figure 2 cells-12-02027-f002:**
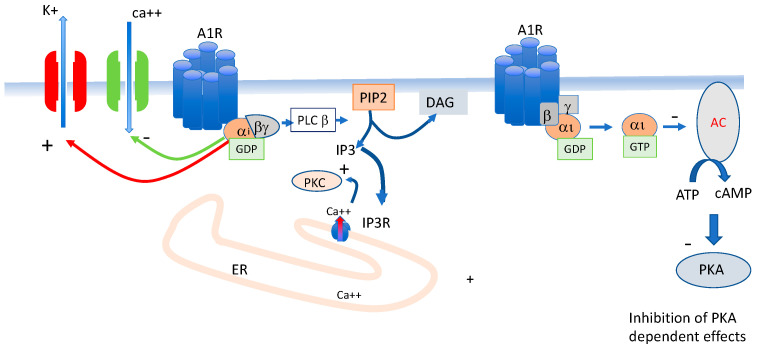
Main consequences of A_1_ R activation on excitable cells.

**Figure 3 cells-12-02027-f003:**
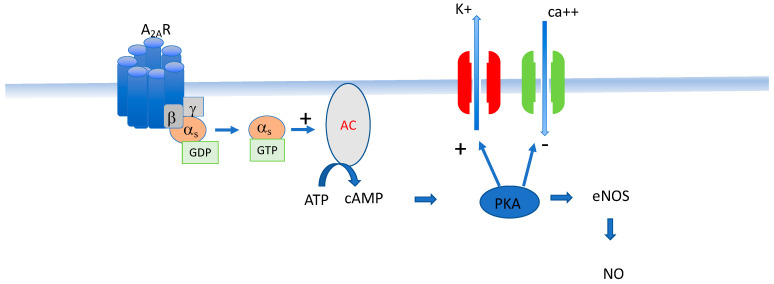
Main consequences of A_2A_ R activation on excitable cells.

**Figure 4 cells-12-02027-f004:**
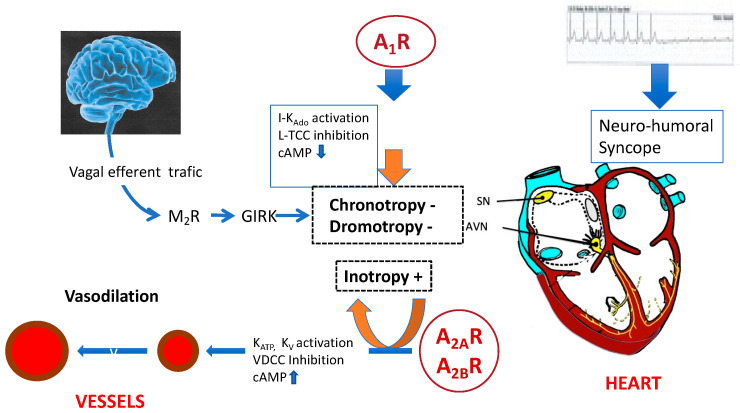
Physiological effects of the activation of adenosine receptors on the cardiovascular system. The main adenosine receptors implicated in NES are the A_1_ and A_2_ subtypes. Activation of A_1_ receptors (A_1_ R) leads to the inhibition of cAMP production (indirect effects), but also to the activation of the inward rectifying K^+^ current (I_. kado/Ach_). This current is also activated by acetylcholine (Ach). A_1_ R activation also leads to the inhibition of L-Type calcium channels (LTCC [16]). The main effect secondary to the activation of A1R is a post-synaptic membrane hyperpolarization of the sinus node (SN) and atrio-ventricular node (AVN) cells, leading to negative chronotropy and dromotropy. These effects are similar to those observed during activation of muscarinic receptors (M2R) and G protein-coupled inwardly rectifying K^+^ channels (GIRK) following vagal stimulation. In the myocardium, activation of adenosine A2 receptors (A2R) leads to inotropy, mostly via indirect (cAMP-dependent) effects. In the vessels, activation of KATP and KV channels and inhibition of voltage-dependent calcium channels (VDCC) leads to smooth cell relaxation and vasodilation.

**Figure 5 cells-12-02027-f005:**
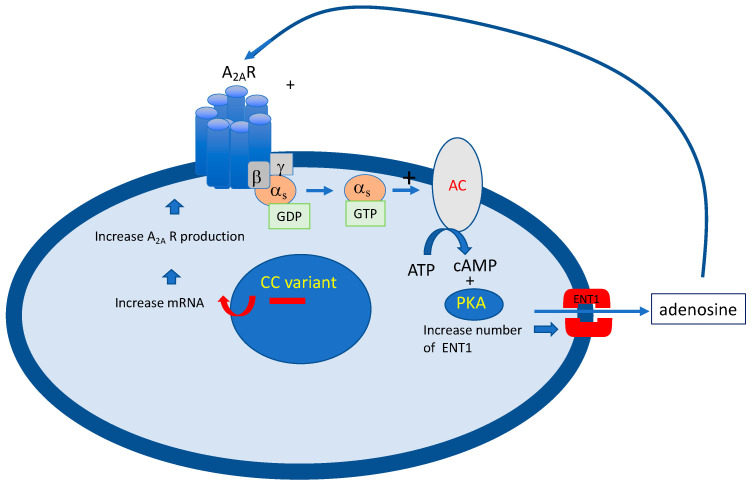
The vicious circle of A2A R.

**Figure 6 cells-12-02027-f006:**
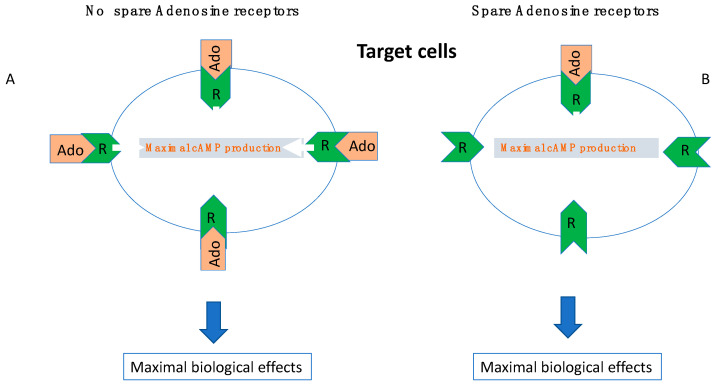
In the absence of adenosine receptor reserve (spare, **A**) maximal biological effects (cAMP production) occur when all the receptors of a target cell are occupied by adenosine (Ado). In the presence of spare receptors; (**B**) maximal biological effects occur even if a weak fraction of adenosine receptors are occupied.

**Figure 7 cells-12-02027-f007:**
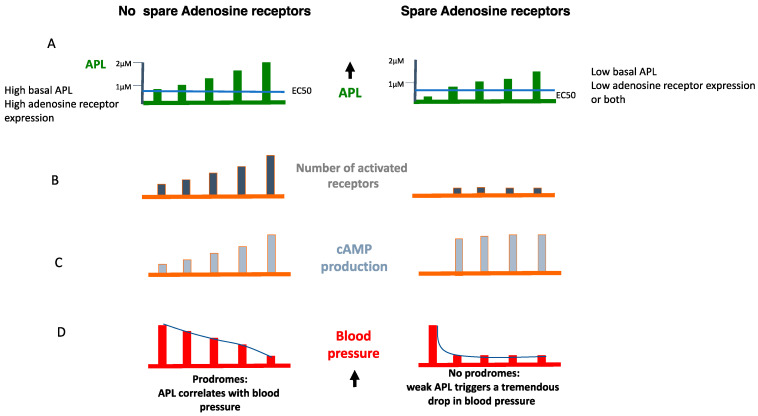
Main consequences of the pharmacological profile of adenosine receptors on the biological and clinical manifestations of NES/. (**A**): In VVS, (**left panel**) the basal adenosine plasma level (APL) is upper than the EC 50 value, while in sudden syncope (low adenosine level, **left panel**) APL in basal condition is under the EC50 value. (**B**): During an increase in APL that occurs as an example during the HUT, the number of activated receptors increase in VVS (**left panel**) while in the case of the presence of spare receptors (**right panel**) No additional receptors are activated in spite of the increase in APL. Consequently, (**C**) the production of cAMP increases as a function of activated receptors (**left panel**) but remains unchanged in the presence of spare receptors (**right panel**). Finally, (**D**) the decrease in SBP occurs progressively during the prodrome period (left panel) but is more dramatic in the presence of spare receptors (**right panel**). The presence of spare receptors seems to be an adaptive mechanism to low APL, low receptor number or both.

**Figure 8 cells-12-02027-f008:**
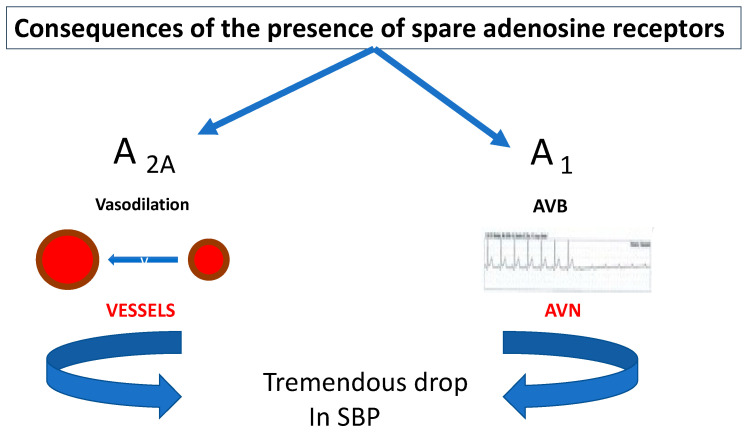
Consequences of the presence of spare A_1_ R or A_2A_ R.

**Figure 9 cells-12-02027-f009:**
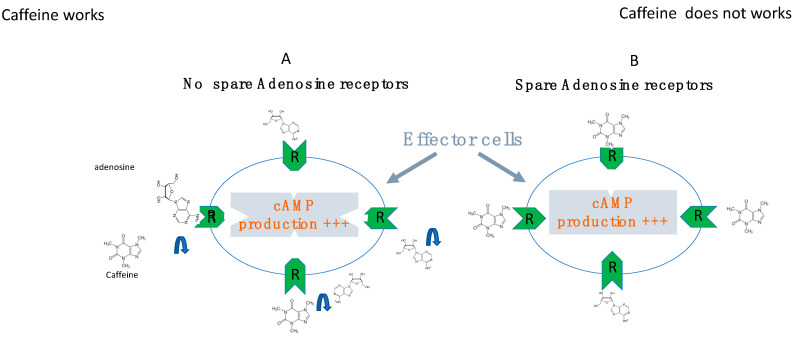
In the absence of reserve receptors (spare receptors, panel **A**), adenosine receptor antagonists competitively displace adenosine from its site and decrease the effects of adenosine on the cardiovascular system in a dose-dependent manner. In the presence of spare receptors (panel **B**), a very small proportion of receptors occupied by adenosine is sufficient to produce maximal biological effects (vasodilation and/or bradycardia). In patients where adenosinemia is low, receptor expression level is low or both, a very small increase in adenosinemia is sufficient to occupy the active fraction of the receptors causing an abrupt drop in blood pressure without prodromes. In these cases, the use of adenosine receptor antagonists such as theophylline or caffeine will be ineffective. Indeed, the concentration of the antagonist would have to displace all the adenosine molecules from its receptors, which requires a very high concentration of the antagonist with side effects that are not tolerated by the patients.

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
