# Peer review of "Adenosinergic System and Neuroendocrine Syncope: What Is the Link?"

_cells, 2023, doi:10.3390/cells12162027_

Round 1
Reviewer 1 Report
This review paper explores the potential role of the adenosinergic system in Neurohumoral Syncope (NHS), also known as reflex syncope. The authors explore meticulously the source and metabolism of adenosine, as well as its impact on the cardiovascular system through its receptors. They also discuss the adenosinergic profile in NHS patients and the role of receptor reserve in clinical presentation. While the authors provide a comprehensive overview of the theoretical background, there are still diagnostic and therapeutic challenges that remain unanswered and could be further elucidated in this review paper.
One area of interest is the potential role of an adenosine test in the diagnostic workup of NHS. Given the involvement of the adenosine pathway in the ADT response, it is worth exploring whether this test could have an additional role to HUTT in certain cases. This is particularly relevant as the most recent ESC guidelines on cardiac pacing have integrated ADT as part of the investigation of patients over 40 years with reflex syncope, raising questions about the usefulness of cardiac pacing.
Another area that warrants further exploration is the measurement of plasma adenosine. It would be useful to report on the published methods of measuring plasma adenosine and the encountered difficulties on adenosine measurement. If there is a reliable, easy, and cheap method for measuring adenosine plasma in syncopal patients, we may be able to select the appropriate therapy as for example use of Theophylline in a more individualized manner.
Apart from low or high adenosine plasma levels in baseline status there are reports that distinguish NHS type according to the pattern of ADP release during HUTT . It would be nice if the authors inlude this information and comment on this data. This would not only enhance the overall quality of the work but also ensure that readers have access to all relevant work.
Finally, the section on clinical implications of Theophylline or other xanthine derivatives could be expanded. While the authors touch on this topic, further exploration of potential treatments and their effectiveness would be beneficial.
Overall, this review paper provides valuable insights into the potential role of the adenosinergic system in NHS. However, further elaboration is needed to fully understand the diagnostic and therapeutic implications of these findings.
Author Response
Reviewer 1
This review paper explores the potential role of the adenosinergic system in Neurohumoral Syncope (NHS), also known as reflex syncope. The authors explore meticulously the source and metabolism of adenosine, as well as its impact on the cardiovascular system through its receptors. They also discuss the adenosinergic profile in NHS patients and the role of receptor reserve in clinical presentation. While the authors provide a comprehensive overview of the theoretical background, there are still diagnostic and therapeutic challenges that remain unanswered and could be further elucidated in this review paper.
One area of interest is the potential role of an adenosine test in the diagnostic workup of NHS. Given the involvement of the adenosine pathway in the ADT response, it is worth exploring whether this test could have an additional role to HUTT in certain cases. This is particularly relevant as the most recent ESC guidelines on cardiac pacing have integrated ADT as part of the investigation of patients over 40 years with reflex syncope, raising questions about the usefulness of cardiac pacing.
Answer
A new paragraph has been added: see 8
- Effects of pacing
In few observational case studies performed in patients with “Low adenosine syncope”, cardiac pacing showed to be successful in preventing syncopal recurrences in patients in whom who had ECG documentation of asystolic pause due to sinus arrest or AV block (3). In a small multicentre trial (57) performed in 80 highly selected elderly patients with unexplained unpredictable syncope who had a positive response to intravenous injection of a bolus of 20 mg of ATP dual-chamber cardiac pacing significantly reduced the 2-year syncope recurrence rate from 69% in the control group to 23% in the active group. For the above reasons, cardiac pacing may be considered to reduce syncope recurrences in patients with the clinical features of adenosine-sensitive syncope according to the most recent ESC guidelines on syncope (56) and on Cardiac pacing (56) (class IIb recommendation).
Another area that warrants further exploration is the measurement of plasma adenosine. It would be useful to report on the published methods of measuring plasma adenosine and the encountered difficulties on adenosine measurement. If there is a reliable, easy, and cheap method for measuring adenosine plasma in syncopal patients, we may be able to select the appropriate therapy as for example use of Theophylline in a more individualized manner.
Answer
A new paragraph has been added intitled: See 4.3 The question of adenosine measurement
4.3 The question of adenosine measurement
Numerous methods have been tried to measure adenosine in blood with varying degrees of success, including high-performance liquid chromatography [33], amperometry [34], and mass spectrometry LC-MS/MS [34-36]. The issue is not so much the assay method as the sampling conditions. The half-life of adenosine is relatively short in blood but longer in a sample tube. Amperometry, which measures adenosine in real time and thus permit kinetic studies, has shown that the half-life of adenosine at room temperature is of the order of 45 s, longer than in circulating blood (unpublished data).A comparative study showed good correlation between high-performance liquid chromatography, amperometry and mass spectrometry (LC-MS/MS), the latter being the gold standard [34]. Adenosine determination after blood collection on blotting paper is convenient because it does not require the use of a stop solution, unlike tube sampling. It also allows easy fingertip capillary sampling and samples can be mailed. Finally the use of blot paper followed by LC-MS/MS appears to be reliable and useful to the dosage of adenosine in blood.
Apart from low or high adenosine plasma levels in baseline status there are reports that distinguish NHS type according to the pattern of ADP release during HUTT . It would be nice if the authors inlude this information and comment on this data. This would not only enhance the overall quality of the work but also ensure that readers have access to all relevant work.
Answer
I suppose this referee mean ADP for adenosine plasma levels?
if so, Fragakis et al have shown that patients with positive HUT exhibit a release of ADP during HUT procedure that may explain the lack of prodromes in this patient's population. Fradakis et al is now cited. See 4.1
Finally, the section on clinical implications of Theophylline or other xanthine derivatives could be expanded. While the authors touch on this topic, further exploration of potential treatments and their effectiveness would be beneficial.
Answer
This part has been improved see 7
Theophylline's efficacy in preventing syncope was first demonstrated on a very small number of patients, with an efficacy that seemed superior to Cardiac Pacing [54]. More recently, due to its effects, theophylline has been successfully used in the prevention of syncope recurrence [12,13]. While theophylline's initial indication seemed to be reserved for syncope with low adenosine, it now appears to be effective in syncope with high or normal adenosine concentration [12]. Thus theophylline administration decreases the number of syncope episodes [12], more particularly in patients affected by low adenosine syncope [13]. Apart from the fact that they are quite often poorly supported, the adenosine receptor antagonists can be ineffective in the presence of spare receptors. Indeed in these case, the concentration of antagonist must be very high in order to displace the adenosine from all the receptors that it occupies (see figure 9).
Overall, this review paper provides valuable insights into the potential role of the adenosinergic system in NHS. However, further elaboration is needed to fully understand the diagnostic and therapeutic implications of these findings.
Thank You
Reviewer 2 Report
Neurohumoral syncope (NHS) is a relatively new term used for specific types of syncope. This review deals with the links between the dysfunction of the adenosinergic system and NHS. In a detailed manner, signal transduction pathways for cAMP production and ion channel effects are described. This is important for the development of new therapeutic approaches to syncope.
The article is written in clear language and is very well-readable. The review is comprehensive and relevant. The figures are illustrative and appropriate.
Suggestions for the authors:
1)In the article, the role of CC polymorphism for the adenosine A2R receptor gene is mentioned as a reason for adenosine overproduction in patients with vasovagal syncope. I suggest using the expression " possible genetic predisposition" instead of " genetic predisposition" because this finding was not entirely confirmed in another study (PACE 2016; 39:330–337). Above mentioned polymorphism was not more frequent in HUT-positive patients compared to controls, although there was an association between gene polymorphism and autonomic nervous activity during HUT measured by the method of heart rate variability. Authors may also consider adding this reference or comment on it.
2) Authors use alternate terms neurohumoral syncope (NHS) and neuroendocrine syncope (NES) probably for the description of the same patient population. Only one term should be used in the article.
Author Response
Neurohumoral syncope (NHS) is a relatively new term used for specific types of syncope. This review deals with the links between the dysfunction of the adenosinergic system and NHS. In a detailed manner, signal transduction pathways for cAMP production and ion channel effects are described. This is important for the development of new therapeutic approaches to syncope.
The article is written in clear language and is very well-readable. The review is comprehensive and relevant. The figures are illustrative and appropriate.
Suggestions for the authors:
1)In the article, the role of CC polymorphism for the adenosine A2R receptor gene is mentioned as a reason for adenosine overproduction in patients with vasovagal syncope. I suggest using the expression " possible genetic predisposition" instead of " genetic predisposition" because this finding was not entirely confirmed in another study (PACE 2016; 39:330–337). Above mentioned polymorphism was not more frequent in HUT-positive patients compared to controls, although there was an association between gene polymorphism and autonomic nervous activity during HUT measured by the method of heart rate variability. Authors may also consider adding this reference or comment on it.
Answer:
Genetic predisposition was replaced by possible genetic predisposition and the Publication by Mitro et al [27] has been added See 4.1
2) Authors use alternate terms neurohumoral syncope (NHS) and neuroendocrine syncope (NES) probably for the description of the same patient population. Only one term should be used in the article.
Answer
We used now Neuroendocrine (NES) Syncope throughout the main body